# The Interactions of the 70 kDa Fragment of Cell Adhesion Molecule L1 with Topoisomerase 1, Peroxisome Proliferator-Activated Receptor γ and NADH Dehydrogenase (Ubiquinone) Flavoprotein 2 Are Involved in Gene Expression and Neuronal L1-Dependent Functions

**DOI:** 10.3390/ijms24032097

**Published:** 2023-01-20

**Authors:** Gabriele Loers, Ralf Kleene, Ute Bork, Melitta Schachner

**Affiliations:** 1Zentrum für Molekulare Neurobiologie, Universitätsklinikum Hamburg-Eppendorf, Martinistr. 52, 20246 Hamburg, Germany; 2Keck Center for Collaborative Neuroscience, Department of Cell Biology and Neuroscience, Rutgers University, 604 Allison Road, Piscataway, NJ 08854, USA

**Keywords:** cell adhesion molecule L1, topoisomerase 1, PPARγ, NDUFV2, Nrxn1, Nlgn1, neurite outgrowth, neuronal survival

## Abstract

The cell adhesion molecule L1 is essential not only for neural development, but also for synaptic functions and regeneration after trauma in adulthood. Abnormalities in L1 functions cause developmental and degenerative disorders. L1’s functions critically depend on proteolysis which underlies dynamic cell interactions and signal transduction. We showed that a 70 kDa fragment (L1-70) supports mitochondrial functions and gene transcription. To gain further insights into L1-70’s functions, we investigated several binding partners. Here we show that L1-70 interacts with topoisomerase 1 (TOP1), peroxisome proliferator-activated receptor γ (PPARγ) and NADH dehydrogenase (ubiquinone) flavoprotein 2 (NDUFV2). TOP1, PPARγ and NDUFV2 siRNAs reduced L1-dependent neurite outgrowth, and the topoisomerase inhibitors topotecan and irinotecan inhibited L1-dependent neurite outgrowth, neuronal survival and migration. In cultured neurons, L1 siRNA reduces the expression levels of the long autism genes neurexin-1 (Nrxn1) and neuroligin-1 (Nlgn1) and of the mitochondrially encoded gene NADH:ubiquinone oxidoreductase core subunit 2 (ND2). In mutant mice lacking L1-70, Nrxn1 and Nlgn1, but not ND2, mRNA levels are reduced. Since L1-70’s interactions with TOP1, PPARγ and NDUFV2 contribute to the expression of two essential long autism genes and regulate important neuronal functions, we propose that L1 may not only ameliorate neurological problems, but also psychiatric dysfunctions.

## 1. Introduction

The cell adhesion molecule L1 regulates important functions in the developing and adult nervous system under physiological and pathological conditions [1,2,3]. In humans, abnormalities in L1 functions are linked to neurological and psychiatric disorders, such as fetal alcohol syndrome, Hirschsprung’s disease, schizophrenia, Alzheimer’s disease, autism and the L1 syndrome [4,5,6,7,8,9,10], which together comprise a group of mild to severe congenital developmental disorders caused by mutations in the L1 gene located on the X chromosome. The spectrum of L1 syndrome disorders includes hydrocephalus, mental retardation, spastic paraplegia, hypoplasia of the corpus callosum, shuffling gait and adducted thumbs [11]. Ablation of L1 in mice leads to malformations and malfunctions of the nervous system [10,11,12,13,14]. In the absence of functional L1, abnormalities occur in neuronal differentiation, migration and survival, axon outgrowth and fasciculation, synaptogenesis, myelination, synaptic plasticity, learning and memory, and behavior as well as regeneration after injury [3,15,16,17,18,19,20,21,22,23,24].

Studies in mice have shown that L1 acts through homophilic and heterophilic interactions and that the proteolytic cleavage of L1 is crucial for its neural functions [24,25,26,27,28,29,30,31]. L1 consists of an extracellular N-terminal part comprising six immunoglobulin-like (Ig) domains and five fibronectin type III (FNIII) domains, a transmembrane portion and a C-terminal intracellular tail [32,33]. L1 can be proteolytically cleaved by different proteases, leading to the generation of soluble and transmembrane fragments [3,15,16,17,18,19,20,21,22,23,24].

Stimulation of L1 signal transduction by the function-triggering L1 antibody 557 generates a transmembrane C-terminal fragment of 70 kDa (L1-70) (Figure 1) via a proteolytically active 60 kDa form of myelin basic protein (MBP), which is released into the extracellular space after stimulation of L1-dependent signal transduction [26]. This form represents a fusion between the 21.5-kDa MBP form and the C-terminal part of dynamin I or a dynamin I-related protein. Notably, L1-70 plays important roles in neuritogenesis, neuronal migration and neuronal survival, as well as in mitochondrial homeostasis [24,26,28,34,35,36].

Another transmembrane C-terminal L1 fragment of 55 kDa (L1-55) is generated by metalloproteases, the β-site of amyloid precursor protein cleaving enzyme and γ-secretase [37,38]. Interestingly, L1-55, but not L1-70, interacts with methyl CpG binding protein 2 (MeCP2) and the heterochromatin protein 1 (HP1) isoforms α, β, and γ and is also involved in L1-dependent functions such as neurite outgrowth and neuronal migration [37,38].

By mass spectrometric analysis upon affinity chromatography using the intracellular domain of L1 (L1-ICD) and a nuclear brain extract we have identified TOP1 as a nuclear molecule that could be associated with L1-ICD [39]. To follow up on this lead, we found that TOP1 may be co-immunoprecipitated with L1 and we propose that L1-70 engages in a complex with TOP1 to regulate the expression of macrophage migration inhibitory factor and to activate microglia [40]. Upregulation of L1-70 expression in a mouse model of Alzheimer’s disease by parabiosis with a wild-type mouse reduced amyloid-β plaques [40]. For this study, we investigated the proposed L1-70/TOP1 interaction in more detail.

**Figure 1 ijms-24-02097-f001:**
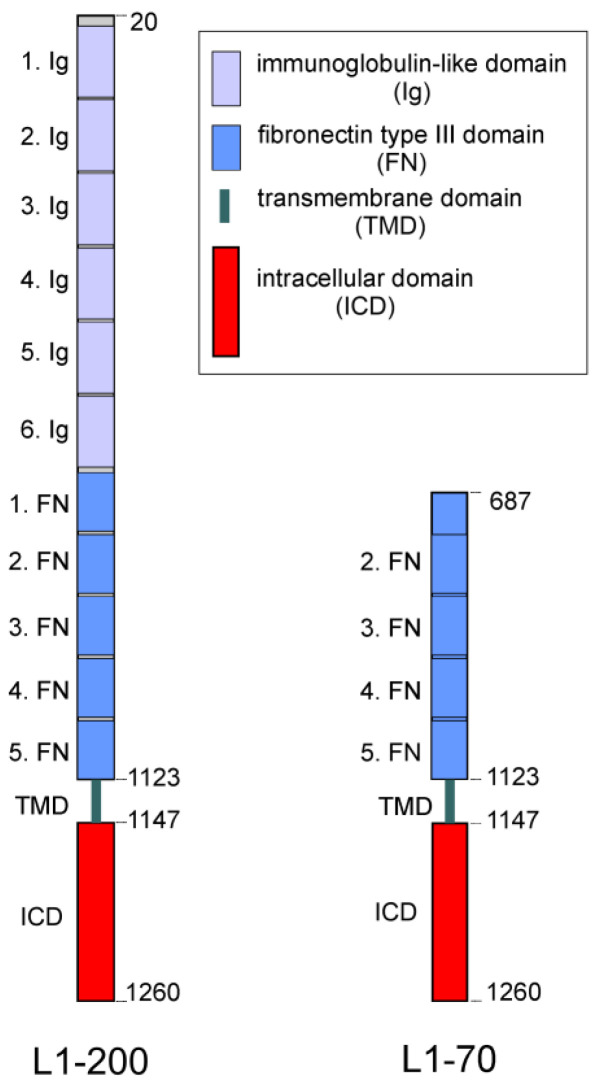
Schematic representation of full-length L1 and L1-70. Full-length L1 (L1-200) consists of an extracellular portion with six Ig-like (Ig) and five FNIII (FN) domains, a transmembrane domain (TMD) and an intracellular domain (ICD). L1-200 is cleaved by a proteolytically active myelin basic protein in the first FNIII domain at position 687, leading to the generation of L1-70 which comprises part of the first FNIII domain, the second, third, fourth and fifth FNIII domains, as well as the TMD and ICD (adapted with permission from [41]; 2023, Kleene, Loers and Schachner).

We also utilized affinity chromatography to find other nuclear and cytoplasmic molecules that could interact directly or indirectly with L1-ICD, including splicing factor proline/glutamine-rich (SFPQ), non-POU domain-containing octamer-binding protein (NonO), paraspeckle component 1 (PSPC1), WD-repeat protein 5 (WDR5), histone H1.4 (HistH1e), nucleoporin 93 kDa (Nup93), heat shock cognate protein 71 kDa (Hsc70), synaptotagmin 1 (SYT1), and importin β1 (impβ1) [39]. Immunoprecipitation experiments using mitochondrial extracts from mouse brains found that NDUFV2 and glyceraldehyde 3-phosphate dehydrogenase (GAPDH) associate with a L1 fragment, and ELISA experiments found that L1-ICD binds directly to NDUFV2 [35]. Immunoprecipitation and ELISA experiments indicated L1’s interaction with nuclear receptors, including PPARγ, retinoid X receptor β (RXRβ), estrogen receptors α (ERα), androgen receptor (AR) and vitamin D receptor (VDR) [36].

For the present study, we investigated whether the aforementioned molecules interact with L1-70. Using immunoprecipitation and proximity ligation assay, we found that TOP1, PPARγ and NDUFV2, but not the other tested molecules, interact with L1-70. Thus, these interactions were chosen for further investigation of their functional consequences.

ELISA showed that the interactions of TOP1 and PPARγ with L1-ICD are mediated via distinct L1-ICD sequences. Reduction of TOP1, PPARγ and NDUFV2 expression by siRNAs or inhibition of topoisomerase activity with topotecan and irinotecan reduced the L1-dependent neurite outgrowth as well as neuronal survival and migration. Reduction of L1 levels by siRNA downregulated the expression levels of the long autism genes Nrxn1 and Nlgn1 and of the mitochondrial gene ND2. Nrxn1 and Nlgn1 mRNA levels were also reduced in mutant mice deficient in L1-70. Our results indicate that the interactions of L1-70 with TOP1, PPARγ and NDUFV2 regulate L1-dependent neuronal functions.

## 2. Results

### 2.1. L1-70 Interacts with TOP1, PPARγ and NDUFV2 via Its Intracellular Domain

Because of the functional importance of the L1-70 fragment [24,26,28,34,35,36], we investigated, by proximity ligation assay, which L1 binding partners interact with L1-70 in a cellular context. Since generation of L1-70 by MBP is blocked by the serine protease inhibitor aprotinin [24,26,28,29], L1-70’s interactions with its binding partners should be reduced by this inhibitor of MBP-dependent L1 proteolysis and L1-70 generation. Thus, to identify predominantly, if not exclusively, L1-70 binding partners, cultured cortical neurons were treated with aprotinin and subjected to proximity ligation with L1 antibodies and antibodies against identified binding partners and not yet critically verified binding partners of L1, including TOP1, SFPQ, NonO, PSPC1, WDR5, HistH1e, Nup93, Hsc70, SYT1, impβ1, ERα, RXR, PPARγ, AR, and VDR. The numbers of L1/TOP1-, L1/PPARγ-, and L1/NDUFV2-positive dots per cell were reduced by aprotinin treatment, whereas aprotinin had no effect on the numbers of L1/SFPQ-, L1/NonO-, L1/WDR5-, L1/PSPC1-, L1/HistH1-, L1/Nup93-, L1/Hsc70-, L1/SYT1-, L1/impβ1-, L1/GAPDH-, L1/ERα-, L1/RXR-, L1/AR-, and L1/VDR-positive dots (Figure 2a,b; Appendix A). The treatment of cortical neurons with function-triggering L1 antibody 557 enhanced the numbers of L1/TOP1-, L1/PPARγ- and L1/NDUFV2-positive dots when compared with non-treated neurons (Figure 2c), indicating that the enhanced generation of L1-70 increases the interaction of TOP1, PPARγ and NDUFV2 with L1-70. Aprotinin treatment reduced the numbers of L1/TOP1-, L1/PPARγ- and L1/NDUFV2-positive dots in unstimulated and L1 antibody 557-stimulated cortical and cerebellar neurons to a similar extent (Figure 2d). Of note, similar results were obtained by the proximity ligation assay when a rabbit L1 antibody and mouse antibodies against TOP1, PPARγ or NDUFV2 were used for proximity ligation. The results indicate that TOP1, PPARγ and NDUFV2 interact with L1-70 in cortical and cerebellar neurons.

Because we were able to show, through ELISA, that NDUFV2 directly binds to L1-ICD [35], we then tested whether TOP1 and PPARγ also bind directly to L1-ICD. A concentration-dependent binding of recombinant L1-ICD to substrate-coated recombinant TOP1 and PPARγ was observed (Figure 3a,b; Appendix A), verifying the direct interaction between L1 and these molecules. To determine the site(s) in L1-ICD that bind(s) to TOP1 and PPARγ, five peptides which cover the total L1-ICD sequence were assayed in a competition ELISA for their capacity to inhibit binding of L1-ICD to TOP1 and PPARγ. NDUFV2 was not analyzed, because we had shown that it binds to the KDET motif in L1-ICD [41]. The peptide P4 comprising L1-ICD amino acids 70–89 strongly reduced the binding of L1-ICD to TOP1 and PPARγ, while peptide P1 comprising amino acids 1–35 moderately reduced the binding of L1-ICD to TOP1 and PPARγ (Figure 3c; Appendix A). Peptides comprising amino acids 23–57 (P2), 55–73 (P3) or 87–114 (P5) did not affect the binding (Figure 3c; Appendix A). These results indicate that amino acids 1–35 and 70–89 are involved in the binding of L1-ICD to TOP1 and PPARγ.

### 2.2. TOP1, PPARγ and NDUFV2 Co-Localizes with L-70 in Cerebellar and Cortical Neurons

We were able to show that L1-70 was not generated in gene-edited mice expressing either L1 with a mutated dibasic sequence in the third FNIII domain at position 858–863 (L1/858–863 mutant; RKHSKR exchanged by SKHSSS) or L1 with a mutation of arginine 687 (L1/687 mutant; R exchanged by A) [25]. Thus, we used these mutants to substantiate the view that L1-70 interacts with TOP1, PPARγ and NDUFV2. Since treatment of neurons with L1 antibody 557 promotes the generation of L1-70 [28], cortical neurons from wild-type and L1 mutant mice were treated with or without this antibody and subjected to proximity ligation assay with a mouse L1 antibody and a goat TOP1 antibody or rabbit antibodies against PPARγ or NDUFV2. The numbers of L1/TOP1-, L1/PPARγ- and L1/NDUFV2-positive dots were similar in non-treated wild-type and mutant cortical neurons (Figure 4a,b). L1 antibody 557-treated wild-type neurons showed enhanced numbers of L1/TOP1-, L1/PPARγ- and L1/NDUFV2-positive dots in comparison to non-treated wild-type neurons, whereas the numbers were not enhanced by the L1 antibody 557 in mutant neurons (Figure 4a,b). These results indicate that TOP1, PPARγ and NDUFV2 interact with the L1-70 that is generated after stimulation of neurons by the L1 antibody 557.

When performing proximity ligation with L1 antibody and antibodies against SFPQ, ERα and RXR, which forms heterodimers with PPARγ [42], we observed that L1 antibody 557 increased the numbers of L1/SFPQ- and L1/ERα-positive dots to the same extent in wild-type and L1/687 mutant cortical neurons (Figure 4a), confirming that SFPQ, ERα and RXR do not interact with L1-70.

To gain more evidence for L1-70’s association with TOP1, PPARγ and NDUFV2, immunoprecipitation was performed with immobilized antibodies and a non-nuclear brain fraction of wild-type and L1-70-lacking mutant mice. Western blot analysis indicates one L1-immunopositive band of 70 kDa in the TOP1, PPARγ and NDUFV2 immunoprecipitates of the non-nuclear wild-type brain fraction, whereas no 70 kDa band was seen in the immunoprecipitates of the non-nuclear brain fractions from the L1 mutants (Figure 4c). No bands were observed when nuclear fractions were used for immunoprecipitation. This result indicates that TOP1, PPARγ and NDUFV2 are associated with L1-70.

### 2.3. Reduction of TOP1, PPARγ and NDUFV2 Expression by siRNAs Reduces L1-Dependent Neurite Outgrowth

To investigate whether the interaction of L1 with TOP1, PPARγ and NDUFV2 plays a role in L1-mediated neurite outgrowth, cortical neurons were mock-transfected or transfected with TOP1, PPARγ and NDUFV2 siRNAs or control siRNA and then treated without and with L1 antibody 557 which induces L1-dependent neurite outgrowth [26,28,29]. Treatment with the antibody 557 promoted neurite outgrowth from mock-transfected neurons (Figure 5a,b). This enhanced neurite outgrowth was prevented by transfection with TOP1 or PPARγ siRNA and reduced by NDUFV2 siRNA (Figure 5a,b), while transfection with control siRNA did not affect the L1-promoted neurite outgrowth (Figure 5a). Neurite outgrowth from non-stimulated neurons was not affected by transfection with TOP1, PPARγ, NDUFV2 or control siRNAs (Figure 5a,b). This result indicates that L1-mediated neurite outgrowth depends on the interaction of L1 with TOP1, PPARγ and NDUFV2.

To analyze whether the reduction of TOP1, PPARγ or NDUFV2 levels by specific siRNAs affects the interaction of L1 with TOP1, PPARγ or NDUFV2, we performed proximity ligation after transfection of cortical neurons with siRNAs. In parallel, we analyzed whether transfection with L1 siRNA also affects these interactions. The numbers of L1/TOP1-, L1/PPARγ- and L1/NDUFV2-positive dots were reduced by more than 60% with the TOP1, PPARγ and NDUFV2 siRNAs as well as L1 siRNA (Figure 6a), indicating that the L1 interactions with TOP1, PPARγ and NDUFV2 are strongly reduced by the siRNAs. Western blot analysis showed reduced TOP1 levels (42.4 ± 8.5%; *n* = 3) in neurons after transfection with the corresponding siRNA relative to the levels in neurons transfected with control siRNA (Figure 6b), confirming the reduction of TOP1 expression in cortical neurons by the corresponding siRNA. Since PPARγ and NDUFV2 were not detectable in cell lysates by Western blot analysis, we determined the PPARγ and NDUFV2 levels by immunofluorescence and observed a reduction of PPARγ and NDUFV2 levels by 57.0 ± 12.4% and 64.1 ± 14.4% (*n* = 3), respectively (Figure 6c).

### 2.4. Disturbance of the L1/TOP1 and L1/PPARγ Interactions by the Cell-Penetrating L1 Peptide P4 Inhibits L1-Dependent Neurite Outgrowth of Cortical Neurons

Because the interaction of L1-ICD with TOP1 and PPARγ was reduced by the L1 peptide P4 (see, Figure 3c), we treated cortical neurons with or without the L1 antibody 557 in the absence or presence of the cell-penetrating tat-P4 peptide in order to analyze whether the interaction of L1 with TOP1 and PPARγ is required for L1-dependent neurite outgrowth in vitro. Treatment with the peptide in the absence of L1 antibody 557 had no effect on neurite outgrowth when compared with untreated neurons (Figure 7a). In the absence of tat-P4 peptide, L1 antibody 557 enhanced neurite outgrowth, while no L1 antibody-enhanced neurite outgrowth was observed in the presence of the peptide (Figure 7a). The results indicate that L1-dependent neurite outgrowth depends on the interaction of L1 with TOP1 and PPARγ and that the disturbance of these interactions reduces L1-stimulated neurite outgrowth.

To investigate whether the tat-P4 peptide interferes with the L1/TOP1 and L1/PPARγ interactions, proximity ligation was performed. In the absence of L1 antibody 557, the peptide reduced the numbers of L1/TOP1- and L1/PPARγ-positive dots by 71.0 ± 11.6% and 75.2 ± 9.6%, respectively (Figure 7b). In the absence of the peptide, L1 antibody 557 treatment increased the numbers of dots per cell relative to untreated neurons, while the numbers were not increased by this antibody in the presence of the peptide (Figure 7b). The peptide reduced the numbers of L1/TOP1- and L1/PPARγ-positive dots in stimulated neurons by 77.4 ± 8.2% and 78.6 ± 10.2%, respectively (Figure 7b). This result indicates that the tat-P4 peptide interferes with the L1/TOP1 and L1/PPARγ interactions.

### 2.5. Inhibition of Topoisomerase Activity Reduces L1-Dependent Neurite Outgrowth, and Neuronal Survival and Migration

Next, we analyzed whether TOP1 activity is required to regulate L1-dependent neurite outgrowth. To this aim, cortical and cerebellar neurons were treated without and with L1 antibody 557 in the absence and presence of the topoisomerase inhibitors topotecan or irinotecan. Neurite outgrowth from non-stimulated neurons was unaffected by the inhibitors, while both inhibitors reduced the L1 antibody-induced neurite outgrowth from cerebellar and cortical neurons (Figure 8a,b).

We also analyzed whether the inhibitors affect neuronal survival after hydrogen peroxide-induced oxidative stress. Hydrogen peroxide treatment induced cell death and this enhanced cell death was reduced in the presence of L1 antibody 557 (Figure 8c,d). The antibody 557-induced cell survival of cerebellar and cortical neurons was reduced by the inhibitors (Figure 8c,d). The inhibitors did not affect hydrogen peroxide-induced cell death in the absence of L1 antibody 557 (Figure 8c,d).

To test whether neuronal L1-dependent migration depends on TOP1 activity, cerebellar explants were incubated with the topoisomerase inhibitors in the absence or presence of L1 antibody 557. The numbers of migrating cells were then determined. Migration in the absence of antibody 557 was not affected by the inhibitors. In the presence of antibody 557, migration was enhanced, and this enhanced migration was prevented by the inhibitors (Figure 8e). The results indicate that L1-mediated neurite outgrowth as well as L1-promoted neuronal survival and migration depends on TOP1 activity.

To analyze whether L1 affects topoisomerase activity, we performed a topoisomerase activity test using nuclear extracts from cortical neurons transfected with or without control siRNA, L1 siRNA or TOP1 siRNA. DNA topoisomerase activity was determined by measuring the extent of the relaxation of superhelical DNA by agarose gel electrophoresis after incubation of nuclear extracts with supercoiled pUC19 plasmid DNA. For control, supercoiled pUC19 plasmid DNA was incubated with reaction buffer instead of nuclear extracts. Compared with the amount of supercoiled DNA after incubation with nuclear extracts from mock control, the amounts of supercoiled DNA were higher after incubation with nuclear extracts from neurons transfected with TOP1 siRNA and the amounts were comparable to those in the buffer control (Figure 8f). In contrast, levels of supercoiled DNA were similar when nuclear extracts from mock, control siRNA and L1 siRNA transfected neurons were incubated with pUC19 plasmid DNA (Figure 8f). This result indicates that transfection of neurons with TOP1 siRNA results in reduced topoisomerase activity in comparison with mock-transfected neurons, while transfection of L1 siRNA or control siRNA did not affect topoisomerase activity. Thus, the reduction of L1 levels does not affect topoisomerase activity, indicating that the interaction of L1 with TOP1 does not regulate the activity of TOP1.

### 2.6. Nrxn1 and Nlgn1 mRNA Levels Are Reduced by L1 siRNA and in L1-70-Deficient Mouse Brains

Because TOP1 and L1 are associated with autism and because TOP1 inhibitors and knockdown of TOP1 reduced the expression of extremely long genes that are associated with autism [10,43,44], we investigated whether the interaction of L1 with TOP1 affects gene expression of autism-associated genes. To this aim, L1 levels were reduced by transfection of cortical neurons with L1 siRNA. As possible targets we chose long genes which are associated with autism, namely Nrxn1 [45,46], Nlgn1 [47,48], contactin-associated protein 2 (Cntnap2) and discs large MAGUK scaffold protein 2 (Dlg2) [49,50]. The levels of Nrxn1 and Nlgn1 mRNAs, but not the levels of Cntnap2 and Dlg-2 mRNAs, were reduced by L1 siRNA when compared with mock-treated neurons (Figure 9a). These results suggest a special relationship between L1 and Nrxn1 and Nlgn1.

Since the mitochondrial TOP1 form, mtTOP1, [51,52] regulates mitochondrial gene expression via relaxation of mitochondrial DNA [51] and since L1 contributes to mitochondrial homeostasis [34], we analyzed whether the L1/TOP1 interaction affects gene expression of the mitochondrially encoded genes NADH:ubiquinone oxidoreductase core subunit 1 (ND1), 2 (ND2), 3 (ND3), 4 (ND4), 4L (ND4L), 5 (ND5), and 6 (ND6) as well as cytochrome b (mt-Cytb), ATP synthase 6 (Atp6), and cytochrome c oxidase I (Cox1), II (Cox2), and III (Cox3). The levels of ND2 mRNA, but not the levels of the other mitochondrial mRNAs, were reduced by L1 siRNA when compared with mock-treated neurons (Figure 9b). These findings suggest a remarkable functional influence of L1/TOP1 interaction on an important mitochondrial protein.

In summary, the results indicate that the L1/TOP1 interaction is required for regulation of Nrxn1, Nlgn1 and ND2 gene expression.

Because TOP1 interacts with L1-70, we considered whether L1-70 might be involved in the regulation of Nrxn1, Nlgn1 and ND2 gene expression. To this aim, the Nrxn1, Nlgn1 and ND2 mRNA were determined in the L1-70-lacking L1/858–863 and L1/687 mutant mice and their wild-type littermates. In comparison with the mRNA levels in cerebellar neurons from wild-type mice, the mRNA levels of Nrxn1 and Nlgn1, but not of ND2, were reduced in cerebellar neurons from mutant mice (Figure 9c). This result indicates that L1-70 is required for the L1-dependent regulation of Nrxn1 and Nlgn1 gene expression.

## 3. Discussion

In previous studies we investigated the proteolysis of full-length L1 and showed that a 70 kDa transmembrane fragment of L1 is generated by cleavage of L1 by MBP (Figure 1) [24]. L1-70 was shown to be transported via the cytoplasm into mitochondria and the nucleus and to interact with mitochondrial and nuclear proteins [28,36,37]. Further studies have indicated that the binding of L1-70 to nuclear and mitochondrial receptors regulates mitochondrial metabolism and trafficking, synaptic plasticity and motor coordination, and promotes neuritogenesis and cell survival [24,26,28,35,36]. Treatment of mouse neuroblastoma N2a cells with the L1 agonist tacrine led to enhanced levels of full-length L1, L1-70 and macrophage migration inhibitory factor [40]. Connection by parabiosis of Alzheimer’s model mice with wild-type mice enhanced hippocampal expression of full-length L1 and L1-70, decreased amyloid β deposition, enhanced expression of macrophage migration inhibitory factor and increased activation of microglia [40]. These results suggest that full-length L1 and L1 fragments can regulate microglial functions. This notion is strengthened by the way in which the coating of microelectrodes with L1 was found to lead to reduced superoxide production by microglia in response to phorbol myristate acetate, as well as the way in which it was found to decrease inducible nitric oxidase, nitric oxide, and pro-inflammatory cytokine levels and increase levels of anti-inflammatory cytokines in lipopolysaccharide stimulated microglia [53]. Furthermore, surface immobilization of full-length L1 on silicone-based neural probes has been found to lead to reduced microglial activation, increased axonal densities and enhanced neuronal survival in the vicinity of the probes [54,55,56]. Since L1 and L1-70 are important for neuronal and microglial functions, and since interactions of L1-70 with intracellular binding partners regulate gene expression and cellular functions, we set out to identify novel L1-70 interaction partners and to study the function of these interactions.

The combined observations of the present study show that TOP1, PPARγ and NDUFV2 are L1-70 binding partners: (1) the interactions of TOP1, PPARγ and NDUFV2 are reduced when the MBP-mediated generation of L1-70 is inhibited by the serine protease inhibitor aprotinin; (2) TOP1, PPARγ and NDUFV2 co-immunoprecipitate L1-70 from brain fractions of wild-type, but not L1-70-lacking mutant mice; and (3) L1 antibody 557-induced generation of L1-70 [26,28] leads to enhanced levels of L1/TOP1, L1/PPARγ and L1/NDUFV2 interactions in wild-type, but not L1-70-lacking neurons. The interactions between L1 and the other tested L1 binding partners, including SFPQ, NonO and GAPDH, are not affected by aprotinin treatment, and L1 antibody 557 treatment enhances the levels of L1/SFPQ, L1/ERα and L1/RXR interactions in wild-type and L1-70-lacking neurons, indicating that these binding partners do not interact with L1-70. SFPQ, NonO and GAPDH have been reported to co-immunoprecipitate with a 70 kDa L1 fragment [35,39]. We assume that this fragment is not identical to L1-70 and represents a different L1 fragment with an identical apparent molecular weight. Since similar levels of L1/TOP1, L1/PPARγ and L1/NDUFV2 interactions are observed in non-stimulated wild-type and L1-70-lacking neurons, we suggest that TOP1, PPARγ and NDUFV2 can also interact with other L1 fragments under conditions that remain to be identified.

We have previously reported that NDUFV2 binds to the KDET motif in L1-ICD [41]. We show here that the binding sites for TOP1 and PPARγ are located in two regions comprising L1-ICD’s amino acids 1–35 and 70–89. These results suggest that L1-70 binds to TOP1, PPARγ and NDUFV2 via sequences in its intracellular domain. A cell-penetrating peptide containing the L1 KDET motif and NDUFV2 siRNA has been reported to reduce L1/NDUFV2 interactions and L1-dependent neurite outgrowth [41]. We now show that a cell-penetrating peptide containing amino acids 70–89 of L1 and TOP1 as well as PPARγ siRNAs reduces the levels of L1/TOP1 and L1/PPARγ interactions and L1-dependent neurite outgrowth. These results imply that the binding of TOP1, PPARγ and NDUFV2 to their binding sites in L1-70’s intracellular domain are crucial for regulation of neuronal functions. The functional importance of the interaction of L1-70 with TOP1 is underscored by the finding that topoisomerase inhibitors reduce L1-dependent neurite outgrowth, neuronal migration, and neuronal survival upon exposure to hydrogen peroxide.

Since TOP1 inhibitors and knockdown of TOP1 have been shown to reduce the expression levels of the Nrxn1 and Nlgn1 genes [10,43,44], and since Nrxn1 and Nlgn1 mRNA levels are reduced in neurons after transfection with L1 siRNA and in brains from L1-70-lacking mice, it is tempting to speculate that the TOP1/L1-70 interaction controls transcription of the Nrxn1 and Nlgn1 genes. Interestingly, L1 siRNA reduces the transcription of the mitochondrial gene ND2, while the ND2 mRNA levels are not altered in brains from L1-70-lacking mice. We thus propose that full-length L1 or an L1 fragment different from L1-70 is required for the regulation ND2 gene expression.

TOP1 is a key transcriptional regulator in postmitotic neurons [43,44] and depletion of TOP1 or acute treatment with the TOP1 inhibitor topotecan resulted in the downregulation of long genes in cultured cortical neurons [43]. A number of these down-regulated long genes are associated with autism, including Nrxn1 and Nlgn1, which contribute to the regulation of synaptic functions [57,58]. Topotecan treatment of cultured cortical neurons suppressed spontaneous synaptic activity and impaired excitatory and inhibitory synapse functions [59]. In addition, conditional depletion of TOP1 in postmitotic excitatory cortical and hippocampal neurons results in downregulation of long genes, neurodegeneration and motor deficit at early postnatal stages [60]. Moreover, mutations in TOP1 are reported to be associated with autism [61,62]. Similarly, L1 has also been associated with autism [10] and to Alzheimer’s disease [6,28] and has been reported to ameliorate some aspects of the Alzheimer’s disease pathology [8,63,64]. In particular, L1-70 and its interaction with TOP1 in nuclei has been proposed to play an important role in the pathogenesis of Alzheimer’s disease [40].

TOP1 has also been found in mitochondria, and TOP1-deficient murine embryonic fibroblasts showed a higher mitochondrial membrane potential, increased expression of mitochondrial genes, including ND2, decreased ATP levels and enhanced levels of reactive oxygen species (ROS) [51]. Because L1-70-lacking mutants showed a reduced mitochondrial membrane potential and reduced ATP levels, it is likely that the interaction between mitochondrial TOP1 and L1-70 controls mitochondrial homeostasis and that an imbalance in this interaction affects the mitochondrial membrane potential, ATP production and ROS generation.

In utero electroporation of NDUFV2 shRNA in the cerebral cortex of mouse embryos resulted in an impaired neuronal migration and accumulation of ectopic neurons in lower cortical layers: most NDUFV2 shRNA-expressing neurons were found in the intermediate and ventricular zones, whereas control cortical neurons migrated into the upper cortical plate [65]. Moreover, most NDUFV2 shRNA-expressing neurons had no or multiple processes and did not develop into polarized neurons, whereas control cortical neurons mostly had bipolar processes and developed a unique axon and several dendrites [65]. These findings imply that NDUFV2 regulates radial migration, development and positioning of cortical neurons. Similar abnormalities in the positioning and morphology of cortical neurons have also been observed in L1-deficient brains [27]. A less compact cortical plate and many misplaced neurons with no or less uniformly orientated dendrites have been observed in cerebral cortices of L1-deficient mouse embryos [27]. Due to the delayed radial migration of L1-deficient cortical neurons, fewer neurons were found in the cortical plate of cerebral cortices from postnatal L1-deficient mice [27]. In addition, impaired migration of L1-deficient cerebellar granule cells and abnormal orientation of neuronal processes in L1-deficient brains have been reported [66,67]. Furthermore, in utero depletion of L1 resulted in delayed neuronal migration [68].

In mice lacking the cytoplasmic polyadenylation element binding protein 1, the expression of NDUFV2 in total brain extracts and the amount of NDUFV2 in mitochondrial complex 1 was reduced [69]. This reduction was accompanied by reduced ATP levels, reduced complex I activity, impaired mitochondrial membrane potential, increased ROS levels and defective dendrite morphogenesis characterized by fewer dendrite branches and reduced dendrite length [69]. L1-70-lacking mice also showed reduced ATP levels, reduced complex I activities, and lower mitochondrial membrane potentials [34] as well as higher ROS levels (unpublished observations). These findings not only support the notion that the interaction of L1-70 with NDUFV2 regulates neuronal migration and positioning of neurons, but also indicate that this interaction plays a role in regulation of mitochondrial homeostasis. Of note, mutations in NDUFV2 have been shown to be responsible for schizophrenia, bipolar disorder, Alzheimer’s and Parkinson׳s disease as well as early-onset hypertrophic cardiomyopathy and encephalopathy [70,71,72,73,74]. Of note, L1 has also been linked to schizophrenia and Alzheimer’s disease [6,8,28,63,64].

PPARγ agonists have been considered for the treatment of Alzheimer’s and Parkinson’s disease [75,76,77,78,79,80,81]. In animal models, PPARγ agonists reduced damage, oxidative stress and neurological deficits and promoted neuronal survival and functional recovery after ischemic brain injury [82,83,84,85]. After spinal cord injury, PPARγ agonists prevented myelin loss, increased proliferation of neuronal precursor cells and improved motor functions [86,87]. It is noteworthy, in this context, that the L1-70 levels were shown to be altered after spinal cord injury or in a mouse model of Alzheimer’s disease, and that L1-70 promoted neurite outgrowth, neuronal cell migration and survival, myelination of axons by Schwann cells in vitro as well as morphological and functional recovery after femoral nerve and spinal cord injury [24,26,28,29]. Small molecule mimetic agonists of L1, overexpression of L1 and application of function-triggering L1 antibodies improved recovery after peripheral nerve and spinal cord injury [19,21,88,89,90,91,92,93,94,95,96,97,98,99,100,101], while blocking L1 with an antagonistic antibody increased numbers of apoptotic neurons and infarct size during the early stage of cerebral ischemia-reperfusion [102]. Moreover, PPARγ agonists attenuated oxidative stress and the behavioral autism-like features in rats treated with valproic or propionic acid, which induce autism-like symptoms, including social impairment, repetitive behavior, hyperactivity, anxiety, and low exploratory activity [103,104]. Treatment of neuron-like cells with PPARγ agonists prevented cell death and increased complex I activity [105,106], and PPARγ agonists modulated mitochondrial fusion–fission events in cultured neurons exposed to oxidative stress [107]. These findings suggest that PPARγ is involved in mitochondrial function and dynamics. Lack of L1-70 affects mitochondrial metabolism and the cellular dynamics characterized by reduced complex I activity, reduced mitochondrial membrane potential, and reduced ATP levels as well as impaired fusion, fission, motility and transport of mitochondria [34,35]. Since PPARγ and L1-70 are involved in regulating mitochondrial function, mitochondrial turnover, energy metabolism, antioxidant defense, and redox balance, it is likely that the interactions between these proteins regulate these mitochondrial functions.

There is increasing evidence that mitochondrial dysfunction and oxidative stress are involved in the pathogenesis of many neurodegenerative diseases and neurodevelopmental disorders, such as autism, attention deficit hyperactivity disorder, schizophrenia, Down’s syndrome and Fragile X syndrome. In particular, abnormalities in the mitochondrial respiratory electron-transport chain during development in utero, infancy, childhood and adulthood may be involved in the etiology of neurodevelopmental disorders and neurodegenerative diseases [108,109]. The mitochondrial respiratory electron-transport chain is responsible for the generation of the mitochondrial membrane potential, of ATP and of ROS and is thus essential for maintaining the physiological function and plasticity of neurons. Abnormalities of the electron-transport chain lead to inhibition of ATP production, accelerated ROS generation, impairments of the energy metabolism and oxidative stress. The disruption of mitochondrial functions then impairs function and plasticity of neurons and may finally lead to the development of neurodevelopmental disorders or neurodegenerative diseases. Based on the combined findings on L1-70, TOP1, PPARγ and NDUFV2, it is conceivable that the functional connections of L1-70 with TOP1, PPARγ and NDUFV2 play important roles under physiological and pathological conditions.

## 4. Materials and Methods

### 4.1. Animals

L1-deficient mice [18] and gene-edited mice expressing L1 with an arginine-to-alanine exchange at position 687 in the first FNIII domain (L1/687 mutant) or with a mutation of the dibasic sequence RKHSKR to SKHSSS at positions 858–863 in the third FNIII domain (L1/858-863 mutant) [25] have been described. Mice were bred and maintained at the Universitätsklinikum Hamburg-Eppendorf at 25 °C on a 12 h light/12 h dark cycle with ad libitum access to food and water. C57BL/6J males and females and L1-deficient and L1 mutant males and their wild-type male littermates were used for all experiments. All animal experiments were conducted in accordance with the German and European Community laws on the protection of experimental animals and approved by the local authorities of the State of Hamburg (animal permit number ORG 1022). The manuscript was prepared following the ARRIVE guidelines for animal research [110].

### 4.2. Reagents and Antibodies

The following antibodies were from Santa Cruz Biotechnology (Dallas, TX, USA): mouse L1 antibody C-2 (NCAM-L1; sc-514360; no RRID available) against the intracellular L1 domain, mouse PPARγ antibody E-8 (sc-7273; RRID:AB_628115), mouse Topo I antibody C-21 (sc-32736; RRID:AB_628382), mouse NDUFV2 antibody F-5 (sc-271620; RRID:AB_10707652), mouse importin β1 (karyopherin β1) antibody H-7 (sc-137016; RRID:AB_2133993), goat TOP1 (Topo I) antibody C-15 (sc-5342; RRID:AB_2205741), goat SYT (synaptotagmin 1) antibody N-19 (sc-7753; RRID:AB_661534), rabbit SFPQ (PSF) antibody H80 (sc-28730; RRID:AB_2186937), rabbit WDR5 antibody H-36 (sc-135245; RRID:AB_10708710), rabbit Nup93 antibody H-300 (sc-292099; RRID:AB_10844043), rabbit AR (androgen receptor) antibody N-20 (sc-816; RRID:AB_1563391), rabbit VDR (vitamin D receptor) antibody C-20 (sc-1008; RRID:AB_632070), rabbit RXR (retinoid X receptors) antibody C-20 (sc-831; RRID:AB_632375), and rabbit ER (estrogen receptor) α antibody HC-20 (sc-543; RRID:AB_631471). Rabbit antibodies against PSPC1 (16714-1-AP; Proteintech Cat# 16714-1-AP, RRID:AB_2878302), NonO (11058-1-AP; Proteintech Cat# 11058-1-AP, RRID:AB_2152167), NDUFV2 (15301-1-AP; RRID:AB_2149048), Hsc70 (10654-1-AP; RRID:AB_2120153), and Histone H1 (18201-1-AP; RRID:AB_10859820) were from ChromoTek and Proteintech Germany (Planegg-Martinsried, Germany). Rabbit antibodies against PPARγ (#2492; RRID:AB_2335662), MeCP2 (D4F3; #3456, RRID:AB_2143849), HP1ɣ (#2619, RRID:AB_2070984), and GAPDH (14C10; #3683, RRID:AB_1642205) were acquired from Cell Signaling Technology Europe (Leiden, The Netherlands). Anti-L1CAM antibody ab12399 against the intracellular L1 domain was acquired from Abcam (Berlin, Germany). Rat monoclonal function-triggering L1 antibody 557 has been described [111]. Secondary antibodies were from Dianova (Hamburg, Germany).

Production and purification of recombinant His-tagged L1-ICD (L1CAM_MOUSE, P11627; amino acids 1147–1260) has been described [112].

Recombinant full-length human NDUFV2 (AR51137PU-S; amino acids 33–249 fused to a His-tag) was from OriGene Technologies (Herford, Germany). Recombinant human DNA Topoisomerase-I (70 kDa, Sf9 insect cells; MBS145206) and recombinant full-length human PPARγ (PPARgamma FL; Cay61700-25) were from Biozol (Eching, Germany).

Synthetic L1 peptides P1 (CFIKRSKGGKYSVKDKEDTQVDSEARPMKDETFGE), P2 (RPMKDETFGEYRSLESDNEEKAFGSSQPSLNGDIK), P3 (GDIKPLGSDDSLADYGGSVD), P4 (CSVDVQFNEDGSFIGQYSGK), P5 (CSGKKEKEAAGGNDSSGATSPINPAVALE) and tat-P4 (YGRKKRRQRRRSVDVQFNEDGSFIGQYSGK) were from Schafer-N (Copenhagen, Denmark). Mouse L1 siRNA (NCAM-L1; sc-43173), mouse TOP1 siRNA (Topo I; sc-36693), mouse PPARγ siRNA (PPARgamma; sc-29456), mouse NDUFV2 siRNA (sc-149892), and control siRNA (Control siRNA-A; sc-37007) were from Santa Cruz Biotechnology. FuGENE^®^ HD Transfection Reagent (Cat #E2311) was from Promega (Walldorf, Germany).

Aprotinin (Cay14716-10), topotecan hydrochloride (Cay14129-10), and irinotecan hydrochloride (Cay14180-25) were from Biomol (Hamburg, Germany) and 4′,6-diamidino-2-phenylindole (DAPI) was from Thermo Fisher Scientific (Darmstadt, Germany).

### 4.3. ELISA

For ELISA, 25 µL of 10 or 20 µg/mL recombinant proteins were incubated overnight at 4 °C in 384-well microtiter plates with high binding surface (Corning, Tewksbury, MA, USA). The following steps were performed at room temperature. Wells were washed with Dulbecco’s phosphate-buffered saline with MgCl_2_ and CaCl_2_ (D8662; Sigma-Aldrich, Taufkirchen, Germany) (PBS), treated with blocking solution (2% essentially fatty acid-free bovine serum albumin in PBS) for 2  h, washed again with PBS containing 0.005% Tween 20 (PBST), and incubated with increasing concentrations of recombinant His-tagged L1-ICD as ligand for 1 h under gentle agitation. For competition ELISA, 2.5 µM L1-ICD were preincubated for 1 h without or with a five-fold molar excess of L1 peptides P1, P2, P3, P4 or P5. The mixtures were then incubated with substrate-coated recombinant proteins. After washing two times with PBS and three times with PBST, L1 antibody C-2 (1:500) in blocking solution was applied for 1 h followed by two washes with PBS and three washes with PBST and incubation with horseradish peroxidase-coupled anti-mouse antibody (diluted 1:2000 in blocking solution) for 1 h. Wells were washed again with PBST and 1 mg/mL ortho-phenylenediamine dihydrochloride (Thermo Fisher Scientific) was used for detection of bound L1-ICD. The reaction was terminated by addition of 25 µL 2.5 M sulphuric acid. Absorbance was measured at 492  nm with an ELISA reader (µQuant; BioTek, Bad Friedrichshall, Germany).

### 4.4. Cultures of Cerebellar and Cortical Neurons and of Cerebellar Explants

Cerebellar neurons were prepared from cerebella of six- to eight-day-old mice. Cerebella were incubated with 10 mg/mL trypsin and 0.5 mg/mL DNase I (Sigma-Aldrich) in Hanks’ balanced salt solution (HBSS) for 15 min at 37 °C, washed with HBSS, mechanically dissociated and centrifuged at 100× *g* for 15 min. Cells were then diluted in Neurobasal A medium (Invitrogen, Darmstadt, Germany) supplemented with 2 mM L-glutamine (Invitrogen), 4 nM L-thyroxine (Sigma-Aldrich), 0.1 mg/mL BSA (Sigma-Aldrich), 12.5 μg/mL insulin (Sigma-Aldrich), 30 nM sodium selenite (Sigma-Aldrich), 100 μg/mL transferrin, 0.1 mg/mL streptomycin and 100 U/mL penicillin (Invitrogen). For the proximity ligation assay, cells were seeded onto poly-L-lysine-coated 12 mm glass coverslips in a 24-well plate at a density of 2.5 × 10^5^ cells per well. For determination of neurite outgrowth, cells were seeded at a density of 5 × 10^4^ cells per well of a 48-well plate coated with poly-L-lysine (Sigma-Aldrich), and for neuronal survival analysis cells were seeded at a density 1.25 × 10^5^ cells per well of a 48-well plate coated with poly-L-lysine. For analysis of RNA levels cells were seeded at a density of 3 × 10^6^ cells per well of a 6-well plate coated with poly-L-lysine.

Cerebellar explants were prepared from six- to seven-day-old mice. Cerebella were forced through Nitrex nets with pore sizes of 300, 200 and 100 μm and tissue pieces were plated on plastic coverslips coated with poly-L-lysine and maintained for 16 h in culture medium containing 20% horse serum. The culture medium was then replaced by serum-free medium.

For culturing of cortical neurons, cerebral cortices were dissected from 15.5- to 16.5-day-old embryos and incubated in 0.025% trypsin (Sigma-Aldrich) in HBSS at 37 °C for 30 min. The cortices were then incubated in HBSS containing 1% BSA (Sigma-Aldrich) and 1% trypsin inhibitor (T-6522, Sigma-Aldrich) at 37 °C for 5 min. After washing in HBSS, the tissue was mechanically dissociated and the dissociated cells were cultured in Neurobasal medium (Invitrogen) supplemented with 1% B-27 (Invitrogen), 2 mM L-glutamine (Invitrogen), 100 U/mL penicillin (Invitrogen) and 100 μg/mL streptomycin (Invitrogen). For the proximity ligation assay and immunostaining, cells were seeded onto poly-L-lysine-coated 12 mm glass coverslips in a 24-well plate at a density of 1.25 × 10^5^ cells per well. For determination of neurite outgrowth, cells were seeded at a density of 5 × 10^4^ cells per well of a 48-well plate coated with poly-L-lysine (Sigma-Aldrich). For Western blot analysis, cells were seeded onto poly-L-lysine-coated 12-well plates at a density of 1.5 × 10^6^ cells per well and for analysis of RNA levels cells were seeded at a density of 3 × 10^6^ cells per well of a 6-well plate coated with poly-L-lysine.

For stimulating L1 functions, neurons were treated with 50 µg/mL L1 antibody 557. For aprotinin treatment, neurons were treated 2 h after seeding with 1 µM aprotinin. Treatment of neurons with 50 µg/mL tat-P4 peptide was performed 30 min after seeding. Treatment with 300 nM topotecan or irinotecan was performed 30 min after seeding (neurite outgrowth) or 30 min before addition of hydrogen peroxide (cell survival). Explants were treated with 300 nM topotecan and irinotecan directly after medium change and then 30 min later with 50 µg/mL L1 antibody 557. Following this, explants were maintained for a further 32 h.

For transfection of cortical neurons, cortical neurons were seeded onto poly-L-lysine-coated coverslips in 24-well plates (proximity ligation and immunostaining), or onto 6-well pates (RNA analysis), 12-well plates (Western blot analysis) and 48-well plates (neurite outgrowth) and maintained for 2 h before transfection with 1 µL (48-well), 2 µL (24-well), 4 µL (12-well) or 8 µL (6-well) 10 µM siRNA and 1–4 µL FuGENE transfection reagent per well. L1 antibody 557 (50 µg/mL) was added to the cultures 24 h after transfection (proximity ligation and cell survival) or 2 h after transfection (neurite outgrowth), and 48 h after transfection the cells were analyzed for neurite outgrowth and cell survival, or used for immunostaining, Western blot analysis, RNA isolation or proximity ligation assay.

### 4.5. Proximity Ligation Assay and Immunostaining with Cerebellar and Cortical Neurons

Cultures were fixed for 15 min at room temperature in 4% formaldehyde, washed with PBS and used for immunostaining or subjected to proximity ligation assay using Duolink PLA products according to the manufacturer’s protocol (Sigma-Aldrich; Duolink PLA technology) with minor modifications. For the proximity ligation assay, cells were incubated with 1% Triton X-100 in PBS for 30 min, washed once with PBS and blocked with Duolink Blocking solution for 30 min before being incubated for 24 h at 4 °C with mouse L1 antibody C-2 and goat or rabbit antibodies against TOP1, SYT, SFPQ, WDR5, Nup93, PSPC1, NonO, NDUFV2, HistH1e, ERα, RXR, PPARγ, AR or VDR or with rabbit L1 antibody ab12399 and mouse antibodies against impβ1, TOP1, PPARγ and NDUFV2, all diluted 1:10 in Duolink Antibody Diluent. Cells were washed twice using Duolink Wash Buffer A and incubated with a mixture of secondary antibodies conjugated with oligonucleotides (Duolink PLA Anti-Rabbit or Anti-Goat Probe MINUS and Duolink Anti-Mouse PLA Probe PLUS). The proximity ligation reaction was then performed according to the manufacturer’s protocol using the Duolink In Situ Detection Reagent RED. Thereafter, the coverslips were incubated in 5 µg DAPI/mL PBS for 15 min, washed twice with PBS and mounted in Immuno-Mount (Thermo Fisher Scientific). Ten images per condition were taken using an Olympus F1000 confocal microscope (Evident Europe GmbH, Hamburg, Germany) and analyzed using ImageJ software (ImageJ, version 1.53; https://imagej.nih.gov/ij/index.html; RRID:SCR_003070, accessed on 8 March 2022). Numbers of red dots and numbers of DAPI-stained nuclei were determined using ImageJ and the number of red dots per image was divided by the number of nuclei per image. The average values of red dots per cell (=nucleus) were determined in ten images per condition.

For immunostaining, fixed cells were incubated with 1% Triton X-100 in PBS for 30 min, washed once with PBS and blocked with Duolink Blocking solution for 30 min before being incubated for 24 h at 4 °C with mouse NDUFV2 and PPARγ antibody diluted 1:10 in Duolink Antibody Diluent. Cells were then washed three times using PBS and incubated with anti-mouse Cy3-conjugated secondary antibody. Afterwards, the coverslips were mounted in Roti-Mount FluroCare DAPI (Carl Roth, Karlsruh, Germany).

### 4.6. Determination of Neurite Outgrowth, Neuronal Migration and Neuronal Survival

Cells and explants were washed gently with pre-warmed culture medium, fixed in 2.5% glutaraldehyde for 30 min at room temperature, and stained with 1% toluidine blue and 1% methylene blue in 1% sodium tetraborate for 30 min at room temperature. Neurite outgrowth and neuronal migration were analyzed by measuring the total length of neurites in an Axiovert microscope with the AxioVision 4.6 imaging system (Carl Zeiss, Oberkochen, Germany) or with Image J counting the number of cells that had migrated out of the explant core. For neurite outgrowth analysis for each condition at least 100 neurons were counted and migrating cells were counted from 12–15 explants per condition.

To determine cell death, neurons were maintained overnight in serum-free medium and then treated with 50 µg/mL of L1 antibody 557 and exposed to oxidative stress by the addition of 10 µM H_2_O_2_ for 24 h. Live and dead cells were then stained with calcein-AM (Thermo Fisher Scientific) and propidium iodide (Sigma-Aldrich) and imaged with a Zeiss AxioObserver.A1 microscope (Carl Zeiss) with a 20× objective (aperture 0.4) and the AxioVision 4.6 software (Carl Zeiss). Live and dead cells were counted in five images (containing 350–400 cells each) from each of three wells per condition and experiment using ImageJ.

### 4.7. Immunorecipitation and Western Blot Analysis

For the preparation of nuclear and non-nuclear fractions, the Subcellular Protein Fractionation Kit for Tissue (Thermo Fisher Scientific) was used. Fractions in cytoplasmic extraction buffer (CEB) and membrane extraction buffer (MEB) were pooled and taken as non-nuclear fraction. Fractions in nuclear extraction buffer (NEB) were taken as nuclear fraction. Nuclear and non-nuclear fractions were used for immunoprecipitation.

For immunoprecipitation, Protein G magnetic beads (25 μL per sample) were washed twice in phosphate-buffered saline (PBS), pH 7.4, and incubated in dilution buffer (1 mg/mL bovine serum albumin in PBS) for 10 min at 4 °C under rotation. The mouse NDUFV2, PPARγ and TOP1 antibodies (10 μg per sample) were diluted in dilution buffer. The diluted antibody solutions were incubated with the beads for 1 h at 4 °C under rotation. The beads were washed in dilution buffer for 5 min at 4 °C under rotation. A freshly prepared 13 mg/mL stock solution of dimethyl pimelimidate (DMP) was 1:1 diluted with wash buffer (0.2 M triethanolamine in PBS). After washing the beads in PBS, the diluted DMP solution (pH 8–9) was added to the beads which were then incubated for 30 min at room temperature (20–24 °C) under rotation. The beads were then washed in wash buffer for 5 min at room temperature under rotation and incubated in DMP solution and wash buffer twice more. Finally, the beads were incubated twice in quenching buffer (50 mM ethanolamine in PBS) for 5 min at room temperature under rotation. After washing, the beads were incubated with brain fractions overnight at 4 °C under rotation. The beads were then washed twice with lysis buffer and once with PBS before being boiled for 5 min in sample buffer (60 mM Tris-HCl, pH 6.8, 2% SDS, 1% β-mercaptoethanol, 6% glycerol, and 0.01% bromophenol blue).

For Western blot analysis, samples were run on 4%–20% Mini-PROTEAN^®^ TGX^™^ Precast Protein Gels (BioRad, Feldkirchen, Germany). PageRuler™ Plus Prestained Protein Ladder (Thermo Fisher Scientific) was used as molecular weight marker. Proteins were then transferred to 0.45 μm Protran™ nitrocellulose membranes (VWR, Darmstadt, Germany) and stained with Ponceau S to control protein loading. The membranes were then incubated for 1 h in blocking solution (5% non-fat dry milk powder in Tris-buffered saline (TBS) (TBS; 10 mM Tris-HCl, pH 7.4; 150 mM NaCl) with 0.05% Tween 20 (TBST) and then incubated overnight with primary antibodies (1:1000) in blocking solution at 4 °C with shaking. After washing five times for 5 min in TBST, the membranes were incubated for 1 h with horseradish peroxidase-conjugated secondary antibody (1:20,000 in blocking solution). Bands were detected using enhanced chemiluminescent solution (ECL Prime and Select Western blotting reagents; GE Healthcare, Solingen, Germany) and a CCD camera (ImageQuant LAS-4000 mini; GE Healthcare).

### 4.8. qRT-PCR Analysis

For reverse transcription, oligoT18 primer and SuperScript^®^ II reverse transcriptase (Thermo Fisher Scientific) was used. qPCR was performed in triplicate using reverse transcribed mRNA, the 7900HT Fast Real-Time PCR System (Thermo Fisher Scientific), the qPCR kit QuantiNova SYBR Green (Qiagen, Hilden, Germany), primers for determination of the mRNA levels of Nrxn1 (fw: AAC GGA CTG ATG CTT CAC ACA; rev: GAT ATT GTC ACC TGA CGC AGA TT), Nlgn1 (fw: AAA CAC AGT GAT TCG CAA GGG; rev: CCT TGG CAC CTA GAT CAC TTG), Cntnap2 (fw: CCT TGG CAC CTA GAT CAC TTG; rev: CCC CTC CAA TGA TAG CTG AGT TT), Dlg2 (fw: ATC ATT CCT TAC CTC GGC TAA CT; rev: TCA GAG GAG AGA TGT GAG ACT G), COX1 (fw: TTT TCA GGC TTC ACC CTA GAT GA; rev: CCT ACG AAT ATG ATG GCG AAG TG), COX2 (fw: TGA AGA CGT CCT CCA CTC ATG A; rev: GCC TGG GAT GGC ATC AGT T), COX3 (fw: GTT TGC CTA CGA CAA CTA AAA TTT C; rev: TGC TGC GGC TTC AAA TCC), CytB (fw: ATT CCT TCA TGT CGG ACG AG; rev: ACT GAG AAG CCC CCT CAA AT), ATP6 (fw: AGC TCA CTT GCC CAC TTC CT; rev: AAG CCG GAC TGC TAA TGC CA), ND1 (fw: CTA GCA GAA ACA AAC CGG GC; rev: CCG GCT GCG TAT TCT ACG TT), ND2 (fw: CCA TTC CAC TTC TGA TTA CC; rev: GTC ATG TAA GAA GAA TAA GTC C), ND3 (fw: TAG TTG CAT TCT GAC TCC CCC A; rev: GAG AAT GGT AGA CGT GCA GAG C), ND4 (fw: CCA GCC TAA CAC TTC TAT G; rev: GGC TAG CTA TTA ATA TTA GTG GC), ND4L (fw: AGC TCC ATA CCA ATC CCC ATC AC; rev: GGA CGT AAT CTG TTC CGT ACG TGT), ND5 (fw: CCT ACT AAT TGG ATG ATG GTA C; rev: CGG TTA TAG AGG ATT GCT TG), ND6 (fw: CTT GAT GGT TTG GGA GAT TGG; rev: ACC CGC AAA CAA AGA TCA CC), and the reference gene actin (fw: TTC TGC ATC CTG TCA GCA ATG; rev: TCC TGT GGC ATC CAT GAA ACT). All primers were from Metabion (Planegg-Steinkirchen, Germany). The SDS 2.4 software (version 2.4.1; Thermo Fisher Scientific) was used for analysis of the qPCR data. The mRNA levels of Nrxn1, Nlgn1, Cntnap2, Dlg2, COX1, COX2, COX3, CytB, ATP6, ND1, ND2, ND3, ND4, ND4L, ND5, and ND6 relative to the mRNA levels of the reference gene actin were calculated. Data of relative gene expression (ΔCt values) were used for statistical analysis.

### 4.9. Preparation of a Nuclear Protein Extract and Topoisomerase Activity Test

Cultured neurons (2 × 10^6^) were scraped off in 300 µL buffer A (10 mM Tris/HCl, pH 7.4, 10 mM NaCl and 1 mM EDTA) and incubated for 15 min on ice. After centrifugation for 8 min at 1000× *g* and 4 °C, the pellet was resuspended in 300 µL buffer B (10 mM Tris-HCl, pH 7.4, 10 mM NaCl and 1.5 mM MgCl_2_), and 100 µL 4 M NaCl was added. An aliquot was saved for determination of the protein concentrations. After incubation for 30 min on ice, the samples were centrifuged for 10 min at 9000× *g* and 4 °C. The supernatant was collected and 400 µL glycerol was added, and the samples were thoroughly mixed and stored as nuclear extracts at −80 °C until use.

In a total volume of 25 µL, nuclear extracts (50–100 ng protein) were incubated with 100 ng supercoiled pUC19 plasmid DNA (Plasmid-DNA pUC19; X911.1; Carl Roth) in reaction buffer (20 mM Tris-HCl, pH 8.1, 1 mM dithiothreitol, 20 mM KCl, 10 mM MgCl_2_, 1 mM EDTA, 30 mg/mL bovine serum albumin) for 30 min at 37 °C. Stopping buffer was added to a final concentration of 1% sodium dodecyl sulfate, 15% glycerol, 50 mM EDTA, pH 8, and 0.5% bromophenol blue, and the samples were subjected to electrophoresis on a 1% agarose gel (89 mM Tris, 89 mM boric acid, 62 mM EDTA; 1 V/cm) and to ethidium bromide staining.

### 4.10. Statistical Analysis

Analyses were performed using SigmaPlot 14.0 (Systat Software GmbH, Düsseldorf, Germany). The types of tests are indicated in the legends. *p*-values of <0.05, <0.01, <0.005 and <0.001 were accepted as a significant difference and indicated by *, **, *** and **** or #, ##, ### and #### or §, §§, §§§ and §§§§.

## Figures and Tables

**Figure 2 ijms-24-02097-f002:**
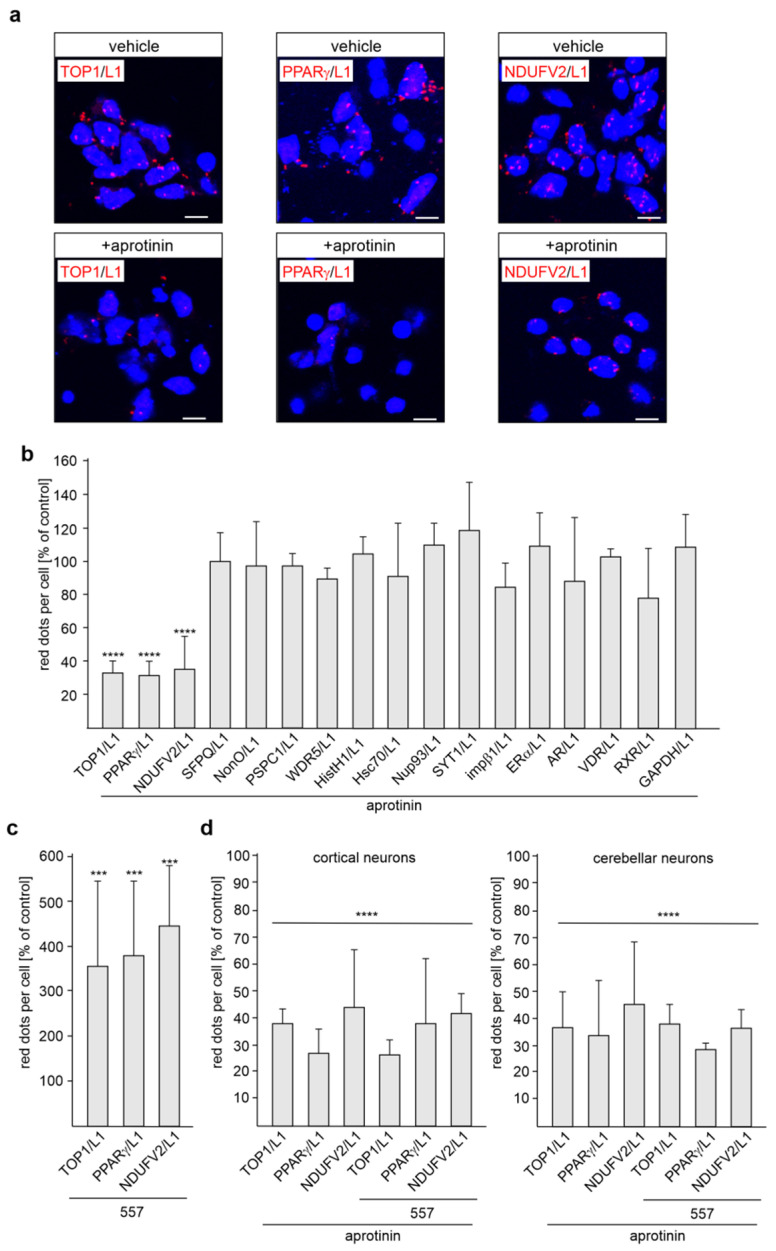
L1-70 interacts with TOP1, PPARγ and NDUFV2, but not with other L1 binding partners in neurons. Cultured cortical neurons were treated with vehicle (-) or with the serine protease inhibitor aprotinin. They were then subjected to proximity ligation with L1 antibody and antibodies against SFPQ, NonO, PSPC1, WDR5, TOP1, HistH1, Nup93, Hsc70, SYT1, impβ1, ERα, RXR, PPARγ, NDUFV2, AR, VDR or GAPDH. (**a**) Representative images of vehicle- and aprotinin-treated neurons stained with the mouse L1 antibody C-2 and rabbit antibodies against TOP1, PPARγ or NDUFV2 are shown. Nuclei are stained with DAPI. Scale bars: 10 µm. (**b**) Mean values + SD from two independent experiments are shown for the average numbers of red dots per cell relative to control (values of vehicle control set to 100%) (**** *p* < 0.001; one-way ANOVA with Bonferroni’s multiple comparison test). (**c**) Cortical neurons were treated with or without L1 antibody 557. Mean values + SD from two independent experiments are shown for the average numbers of red dots per cell relative to control (values of treatment without antibody 557 set to 100%). (*** *p* < 0.005; one-way ANOVA with Bonferroni’s multiple comparison test). (**d**) Cortical and cerebellar neurons pretreated with vehicle or aprotinin were incubated without or with antibody 557 in the absence or presence of aprotinin. Mean values + SD from two independent experiments are shown for the average numbers of red dots per cell relative to control (values of treatments without aprotinin set to 100%) (**** *p* < 0.001; one-way ANOVA with Bonferroni’s multiple comparison test).

**Figure 3 ijms-24-02097-f003:**
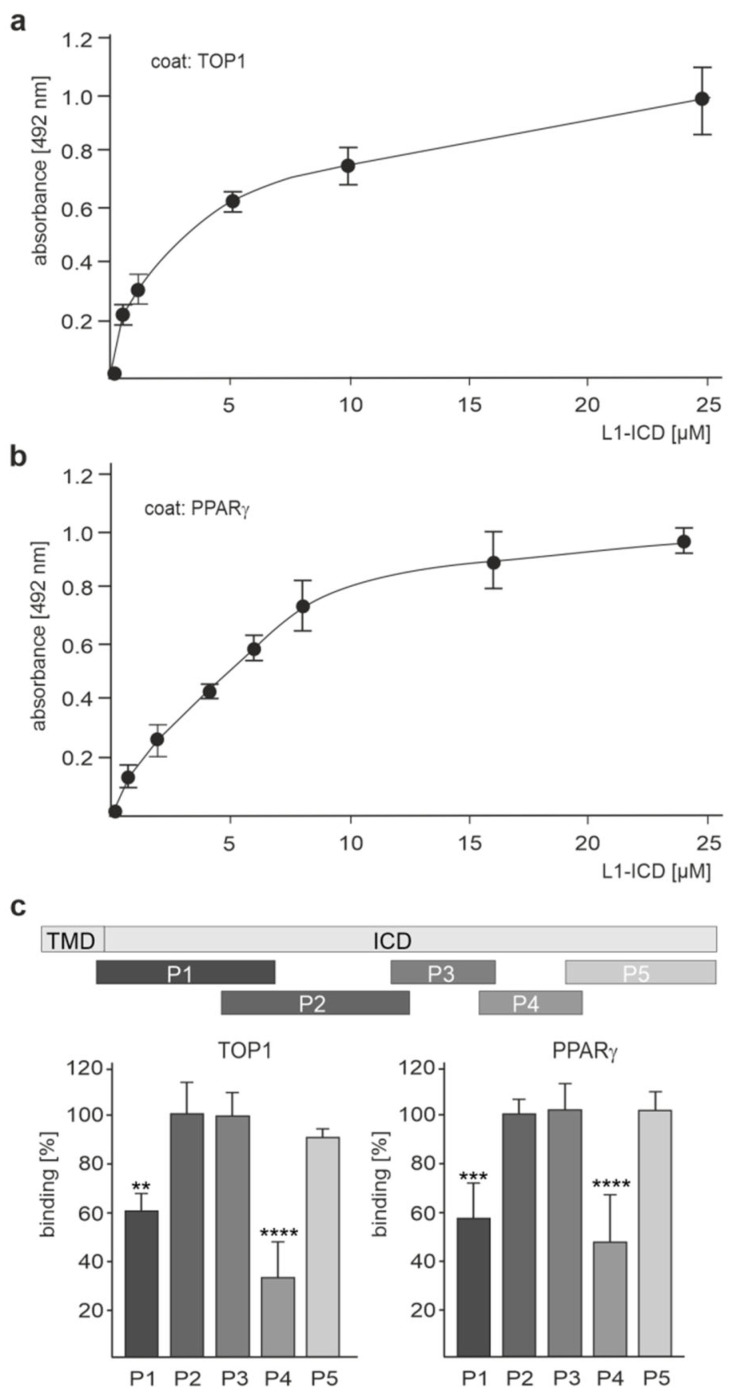
L1-ICD binds to TOP1 and PPARγ. Recombinant TOP1 (**a**) and PPARγ (**b**) were substrate-coated and incubated with increasing concentrations of L1-ICD. Binding of L1-ICD was determined by ELISA using mouse L1 antibody C-2 and horseradish peroxidase-conjugated secondary antibody. Mean values ± SD from three independent experiments carried out in triplicate are shown. (**c**) Recombinant TOP1 and PPARγ were substrate-coated and incubated with L1-ICD in the absence or presence of a five-fold excess of L1 peptides P1, P2, P3, P4 or P5, which cover the entire L1-ICD sequence. The ICD and transmembrane domain (TMD) of L1 and the positions of the peptides in L1-ICD are visualized. Binding was determined by ELISA using mouse L1 antibody C-2 and horseradish peroxidase-conjugated secondary antibody. Mean values + SD from five independent experiments carried out in triplicate are shown for the binding relative to control (values in the absence of peptides set to 100%) (** *p* < 0.01, *** *p* < 0.005, **** *p* < 0.001; one-way ANOVA with Bonferroni’s multiple comparison test).

**Figure 4 ijms-24-02097-f004:**
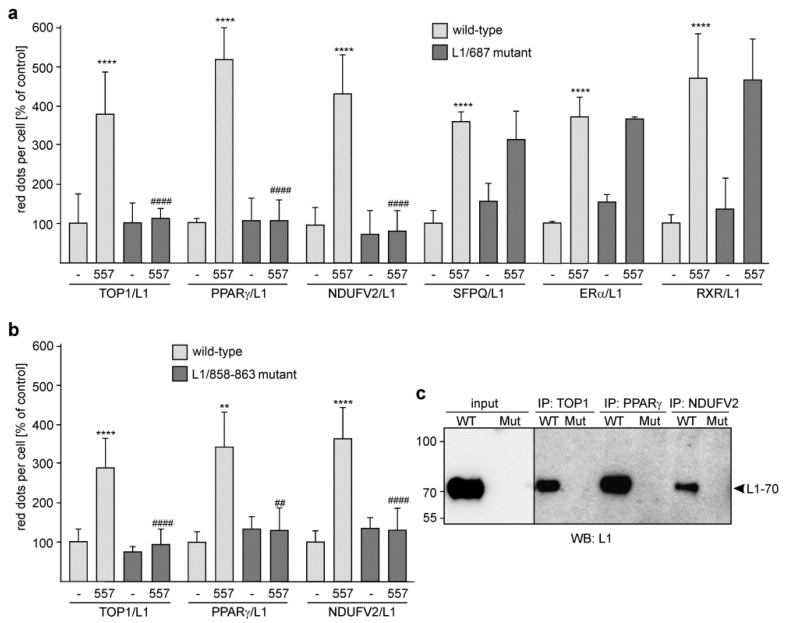
No enhanced interaction of L1 with TOP1, PPARγ and NDUFV2 in stimulated cortical neurons from L1-70-lacking mutant mice. Cultured cortical neurons from L1/687 (**a**) and L1/858–863 (**b**) mutant mice and their wild-type littermates (**a**,**b**) were treated without (-) or with L1 antibody 557 (557) and then subjected to proximity ligation assay with L1 antibody and TOP1, PPARγ or NDUFV2 antibodies. (**a**) For control, the L1 antibody and an antibody against SFPQ, ERα or RXR were used for proximity ligation. (**a**,**b**) Mean values + SD are shown for the average numbers of L1/TOP1-, L1/NDUFV2-, L1/PPARγ-, L1/SFPQ-, L1/ERα-, and L1/RXR-positive dots per cell relative to control (values of unstimulated wild-type neurons set to 100%) (** *p* < 0.01, **** *p* < 0.001 relative to non-stimulated wild-type neurons, ## *p* < 0.01, #### *p* < 0.001 relative to stimulated wild-type neurons; one-way ANOVA with Dunn’s multiple comparison test). (**c**) Non-nuclear brain fractions from wild-type (WT) and L1/858–863 mutant (Mut) mice were subjected to immunoprecipitation (IP) with immobilized mouse antibodies against TOP1, PPARγ and NDUFV2. Non-nuclear fractions (input) and immunoprecipitates were subjected to Western blot (WB) analysis with L1 antibody C-2. The arrowhead indicates L1-70.

**Figure 5 ijms-24-02097-f005:**
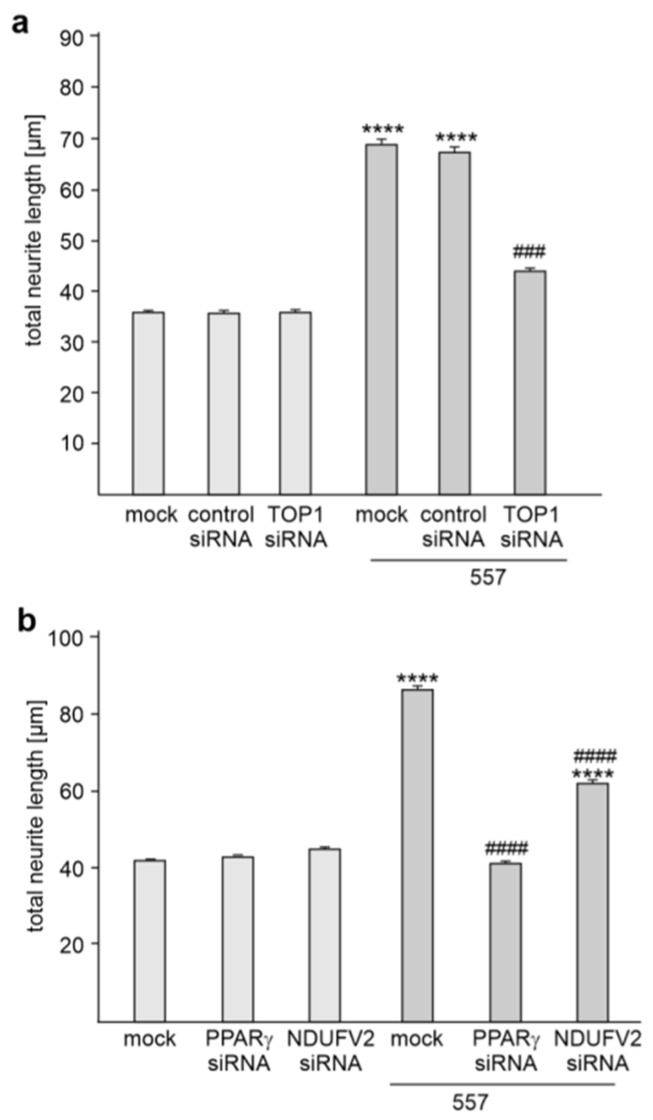
Reduction of TOP1, PPARγ or NDUFV2 expression inhibits L1-dependent neurite outgrowth. (**a**,**b**) Cortical neurons were mock-transfected or transfected with control siRNA, TOP1 siRNA, PPARγ siRNA or NDUFV2 siRNA. Neurons were then treated without or with L1 antibody 557. (**a**,**b**) Mean values + SEM from three independent experiments are shown for total neurite lengths (**** *p* < 0.001 relative to mock-transfected non-stimulated neurons, ### *p* < 0.005, #### *p* < 0.001 relative to mock-transfected stimulated neurons; one-way ANOVA with Dunn’s multiple comparison test).

**Figure 6 ijms-24-02097-f006:**
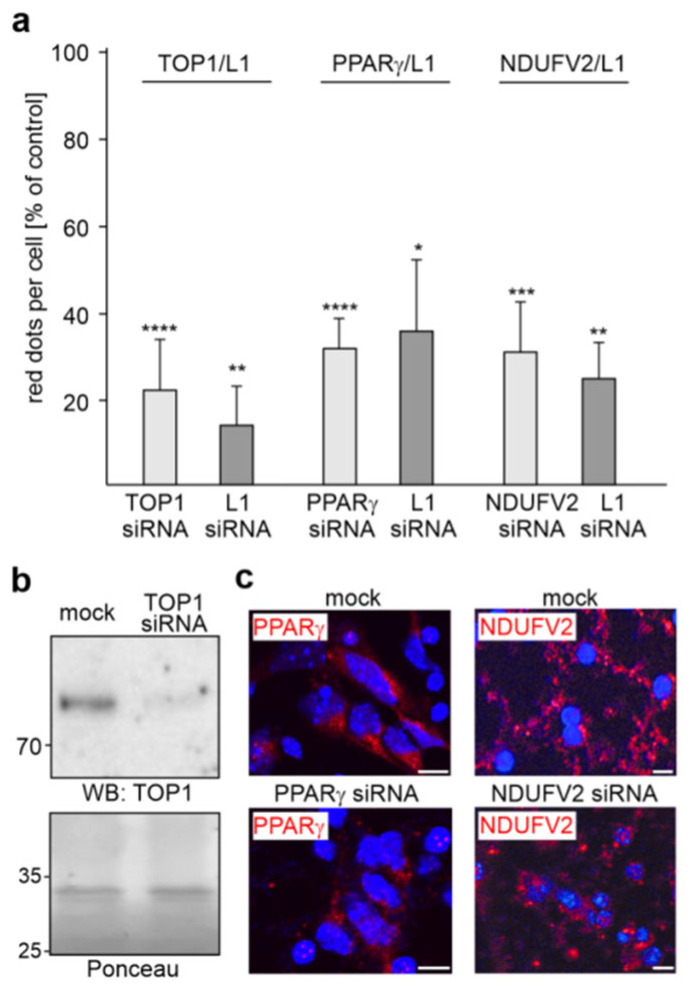
The interaction of L1 with TOP1, PPARγ or NDUFV2 is reduced by L1, TOP1, PPARγ and NDUFV2 siRNAs. (**a**) Cultured cortical neurons were mock-transfected or transfected with TOP1, PPARγ, NDUFV2 or L1 siRNAs. The transfected cells were then subjected to proximity ligation with L1 antibody and antibodies against TOP1, PPARγ or NDUFV2. Mean values + SD from two experiments are shown for the average numbers of L1/TOP1-, L1/PPARγ- and L1/NDUFV2-positive red dots per cell relative to control (values of mock-transfected neurons set to 100%) (* *p* < 0.05, ** *p* < 0.01, *** *p* < 0.005, **** *p* < 0.001; one-way ANOVA with Bonferroni’s multiple comparison test). (**b**) Western blot analysis of lysates from mock-transfected neurons or neurons transfected with TOP1 siRNA using TOP1 antibody. Ponceau S staining of a prominent 35 kDa band served as loading control. (**c**) Mock-transfected neurons or neurons transfected with PPARγ or NDUFV2 siRNAs were subjected to immunostaining with PPARγ or NDUFV2 antibody. Nuclei are stained with DAPI. Representative images are shown and indicate reduced PPARγ or NDUFV2 levels in neurons transfected with PPARγ siRNA or NDUFV2 siRNA. Scale bars: 10 µm.

**Figure 7 ijms-24-02097-f007:**
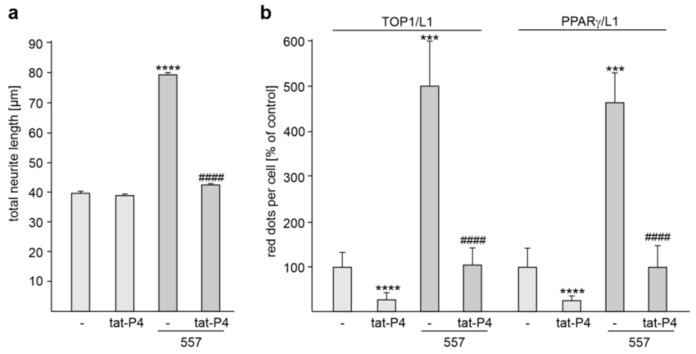
Disturbance of the L1/TOP1 and L1/PPARγ interactions inhibits L1-dependent neurite outgrowth from cortical neurons. (**a**,**b**) Cortical neurons were treated without (-) or with tat-P4 peptide. Neurons were then treated without or with L1 antibody 557 (557). (**a**) Mean values + SEM from three independent experiments are shown for total neurite lengths (**** *p* < 0.001 relative to mock-transfected non-stimulated neurons, #### *p* < 0.001 relative to mock-transfected stimulated neurons; one-way ANOVA with Dunn’s multiple comparison test). (**b**) After the treatments, neurons were subjected to proximity ligation with L1 antibody and TOP1 antibody or PPARγ antibody. Mean values + SD from three independent experiments are shown for average numbers of L1/TOP1- and L1/PPARγ-positive red dots per cell relative to control (values of non-transfected neurons set to 100%) (*** *p* < 0.005, **** *p* < 0.001 relative to mock-transfected stimulated neurons, #### *p* < 0.001 relative to mock-transfected stimulated neurons; one-way ANOVA with Bonferroni’s multiple comparison test).

**Figure 8 ijms-24-02097-f008:**
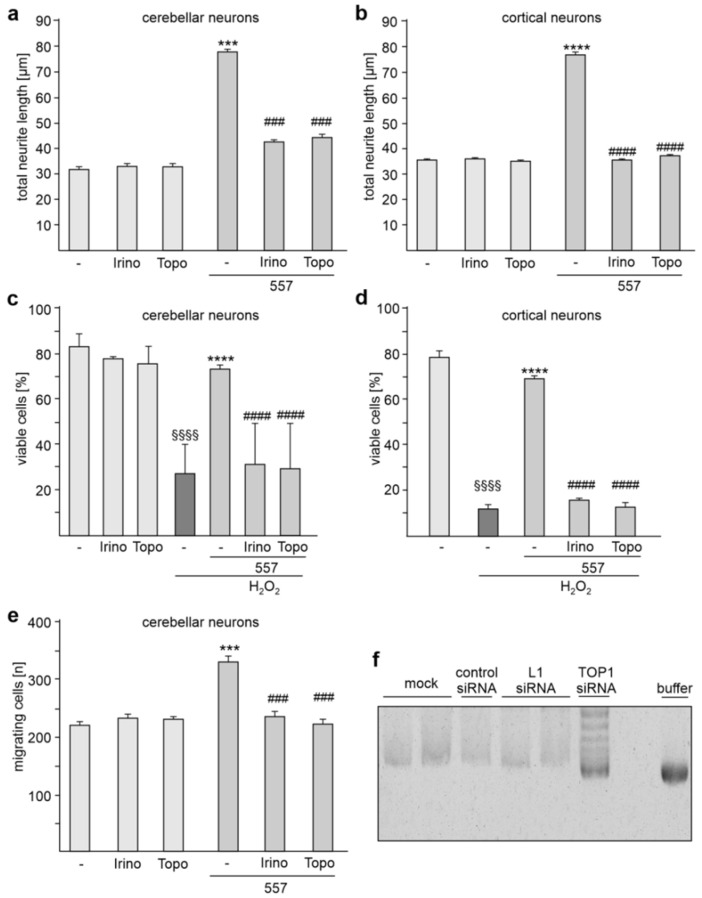
Topoisomerase inhibitors reduce L1-dependent neurite outgrowth as well as neuronal survival and migration. Cerebellar neurons (**a**,**c**) and cerebellar explants (**e**), and cortical neurons (**b**,**d**) were treated without (-) or with topotecan (Topo) or irinotecan (Irino) in the absence or presence of L1 antibody 557. (**a**,**b**) Mean values + SEM from three independent experiments are shown for total neurite lengths (*** *p* < 0.005, **** *p* < 0.001 relative to non-stimulated neurons in the absence of inhibitors, ### *p* < 0.005, #### *p* < 0.001 relative to stimulated neurons in the absence of inhibitors; one-way ANOVA with Dunn’s multiple comparison test). (**c**,**d**) Mean values + SEM from three independent experiments are shown for the numbers of viable cells relative to control (values of unstimulated neurons in the absence of H_2_O_2_ set to 100%) (#### *p* < 0.0001 relative to stimulated neurons in absence of inhibitors and the presence of H_2_O_2_, **** *p* < 0.0001 relative to non-stimulated neurons in absence of inhibitors and the presence of H_2_O_2_, §§§§ *p* < 0.0001 relative to non-stimulated neurons in absence of inhibitors and H_2_O_2_; one-way ANOVA with Dunn’s multiple comparison test). (**e**) Mean values + SEM from three independent experiments are shown for the total numbers of migrating cells (### *p* < 0.005 relative to stimulated neurons in absence of inhibitors, *** *p* < 0.005 relative to non-stimulated neurons in absence of inhibitors; one-way ANOVA with Dunn’s multiple comparison test). (**f**) Nuclear extracts from cortical neurons transfected without (mock) or with control siRNA, L1 siRNA or TOP1 siRNA were incubated with supercoiled pUC19 plasmid DNA. For control, the vector was incubated with the reaction buffer. The samples were then subjected to agarose gel electrophoresis and ethidium bromide staining. A representative image of a stained gel is shown.

**Figure 9 ijms-24-02097-f009:**
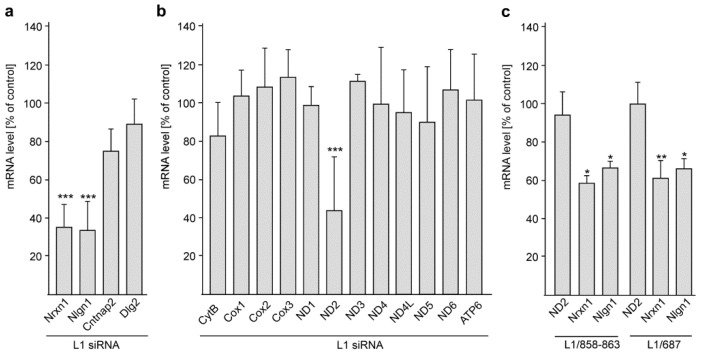
The L1/TOP1 interaction regulates the expression of the long autism genes Nrxn1 and Nlgn1 and the expression of the mitochondrial gene ND2. Cortical neurons were mock-transfected or transfected with L1 siRNA and subjected to RNA isolation and RT-qPCR 48 h after seeding. The mRNA levels of long autism (**a**) or mitochondrial (**b**) genes were normalized to the actin mRNA levels. (**c**) Cerebellar neurons from L1/858-863 mutant, L1/687 mutant and their wild-type littermates were maintained in culture for 48 h and subjected to RNA isolation and RT-qPCR. The ND2, Nrx1 and Nrx1 mRNA levels were normalized to the actin mRNA levels. Mean values + SD from five independent experiments with triplicates (*n* = 5 mice) (**a**,**b**) or from six cultures per genotype (*n* = 6 mice) (**c**) are shown for normalized mRNA levels in neurons after transfection with L1 siRNA relative to the levels in mock-transfected neurons (**a**,**b**) or in mutant neurons relative to wild-type neurons (**c**) (* *p* < 0.05, ** *p* < 0.01, *** *p* < 0.005; one-way ANOVA with Tukey’s (**a**,**b**) or Bonferroni’s (**c**) multiple comparison tests).

## Data Availability

Not applicable.

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
