# Peer review of "The Interactions of the 70 kDa Fragment of Cell Adhesion Molecule L1 with Topoisomerase 1, Peroxisome Proliferator-Activated Receptor γ and NADH Dehydrogenase (Ubiquinone) Flavoprotein 2 Are Involved in Gene Expression and Neuronal L1-Dependent Functions"

_ijms, 2023, doi:10.3390/ijms24032097_

Round 1
Reviewer 1 Report
Presented manuscript is a solid study that sufficiently addresses questions regarding interactions between the cell adhesion molecule L1 and TOP1, PPARy an NDUFV2. Quality of the paper is high and it is a well designed set of experiments. The only consideration I have is to suggest to authors to provide raw ELISA results and their calculations in the supplementary data file as the way they present them in the paper is rare. It does not affect the soundness of the paper, but a more detailed supplementary part is necessary for data audit by readers. My overall comment is to accept this paper in its original version and to add ELISA raw data in the supplementary file.
Author Response
Reviewer 1
Presented manuscript is a solid study that sufficiently addresses questions regarding interactions between the cell adhesion molecule L1 and TOP1, PPARy an NDUFV2. Quality of the paper is high and it is a well designed set of experiments. The only consideration I have is to suggest to authors to provide raw ELISA results and their calculations in the supplementary data file as the way they present them in the paper is rare. It does not affect the soundness of the paper, but a more detailed supplementary part is necessary for data audit by readers. My overall comment is to accept this paper in its original version and to add ELISA raw data in the supplementary file.
Our response: We thank the reviewer for these comments and now provide the raw ELISA data (Supplementary Tables 1-3).
Please also see attachment.

Reviewer 2 Report
Comments are attached

Author Response
Reviewer 2
Comments on Manuscript ID ijms-2134162
Title The interactions of the 70 kDa fragment of cell adhesion molecule L1 with topoisomerase 1, peroxisome proliferator-activated receptor γ and NADH dehydrogenase (ubiquinone) flavoprotein 2 are involved in gene expression and neuronal L1-dependent functions
Authors Gabriele Loers et al
Comments: The authors have carried out novel research on 70kDa Fragment L1 cell adhesion and its functional interactions with TOP1, PPARγ or NDUFV2. There are some point need to be address:
- In Introduction section, line 2, the phrase within the bracket “ (for reviews and references, see)” is not required and needs to be remove from the text.
Our response: We have removed the indicated phrase.
- The authors need to provide the sketch cartoon/ diagram figure of Full length L1 cell adhesion molecule/gene expressing 70kDa Fragment comprising of extracellular domain, transmembrane and Intracellular domain with the location of 70kDa Fragment.
Our response: We now provide a diagram figure showing full-length L1 and the L1-70 fragment (Figure 1).
- What are the fate of The 70kDa Fragment of L1 gene as transmembrane soluble fragment
Our response: We are sorry that we do not understand exactly what the reviewer means by ‘fate’ and can only now mention in the Discussion section the location and functions of L1-70 which we determined in previous studies. We hope that we interpreted the slightly unusual word ‘fate’ to the reviewer’s satisfaction.
- How the 70kDa Fragment of L1 impact the microglial growth and activation process.
Our response: This is an interesting question, but very difficult to answer now, because too little is known about the impact of L1-70 on microglia. The little that is known is now mentioned extremely cautiously in the Discussion section. We would be grateful for the reviewer’s understanding not to go deeper into this not extensively investigated issue.
- How does 70kDa Fragment of L1 influence the EGFR Family members specifically HER4.
Our response: Since we did not investigate if L1-70 binds to EGFR family members or influences EGFR signaling and since there is no literature on L1 influencing or interacting with HER4, we do not want to speculate on this. We think that we should not mention these points in the Discussion section, since other receptor families could come to mind and raise questions as to the influences on other receptor families.
- In the result section, Figure 1a, authors provided the representative images of vehicle- and aprotinin-treated neurons against TOP1, PPARγ or NDUFV2 which were significantly changes but needs to provide the rest of unchanged representatives which were provided in the bar diagram 1b.
Our response: We have added the representative images for the proximity ligation assay of SFPQ/L1, NonO/L1, PSPC1/L1, WDR5/L1, HistH1/L1, Hsc70/L1, Nup93/L1, SYT1/L1, impβ/L1, ERα/L1, AR/L1, VDR/L1, RXR/L1 and GAPDH/L1 (Supplementary Figure 1).
Please also see attachment.
